# Association between Peptic Ulcer Disease and Osteoporosis: The Population-Based Longitudinal Cohort Study in Korea

**DOI:** 10.3390/ijerph16152777

**Published:** 2019-08-03

**Authors:** Phill Hoon Yoon, Sang Joon An, Seok-Hoo Jeong, Yun-Jung Yang, Yeon-pyo Hong

**Affiliations:** 1Division of Gastroenterology, Department of Internal Medicine, Catholic Kwandong University International St. Mary’s Hospital, Incheon 22711, Korea; 2Department of Neurology, Catholic Kwandong University International St. Mary’s Hospital, Incheon 22711, Korea; 3Institute of Biomedical Science, Catholic Kwandong University International St. Mary’s Hospital, Incheon 22711, Korea; 4Department of Preventive Medicine, College of Medicine, Chung-Ang University, Seoul 06974, Korea

**Keywords:** peptic ulcer, osteoporosis, cohort studies, calcium

## Abstract

Objectives: The association between peptic ulcer disease (PUD) and osteoporosis remains unclear. We investigated the association between PUD and osteoporosis by classifying individuals based on gender in a prospective study on 10,030 adults in Korea at a 12-year follow-up examination. Design and setting: The baseline survey of the Ansung–Ansan cohort studies was conducted from May 2001 to February 2003, and 10,030 participants (5018 from the Ansung study and 5012 from the Ansan study) completed the examination. Primary outcome measures: The risk of developing osteoporosis was higher in both men and women in the PUD group than in the control group. Results: At the 12-year follow-up, osteoporosis had developed in 11.1% (21/189) and 29.9% (56/187) of men and women in the PUD group, respectively. Meanwhile, in the control group, 4.8% (70/1464) and 16.5% (271/1639) of men and women, respectively, were presented with osteoporosis. The incidence rates per 1000 person–years were 20.5% and 68.5% in men and women in the PUD group and 11.2% and 42.3% of men and women in the control group, respectively. The risk of developing osteoporosis was higher in both men and women in the PUD group than in the control group (men: hazard ratio [HR] = 1.72, 95% confidence interval [CI] = 1.02–2.92; women: HR = 1.62, 95% CI = 1.20–2.18). Conclusion: The risk of developing osteoporosis was significantly higher in both men and women in the PUD group than in the control group.

## 1. Strengths and Limitations of this Study

►This is a prospective study on a large cohort of participants, with a 12-year follow-up examination conducted.►This study has considered age, hormone replacement therapy, and menopause in women to minimized confounding factors. ►This paper lacks information on the major factors of PUD such as *H. pylori* infection or the use of NSAIDs, which are study limitations.

## 2. Introduction

Peptic ulcer disease (PUD) is a condition affecting the digestive system, which is characterized by a disequilibrium between protective attributes, such as prostaglandin, bicarbonate, and mucus, and damaging attributes, including gastric acid, bile salt, and pepsin in the gastrointestinal mucosa [1]. PUD is classified into two types: Gastric and duodenal ulcer. PUD is caused by *Helicobacter pylori* infection, the use of nonsteroidal anti-inflammatory drugs (NSAIDs) and acetylsalicylic acid, smoking, and alcohol drinking [2,3]. PUD causes complications such as gastrointestinal bleeding, obstruction, and perforation, which increase mortality risk [4]. Therefore, the complications of PUD may substantially increase socioeconomic burden. 

Osteoporosis is a skeletal disease characterized by weak bones, which is determined by assessing bone mineral density and bone quality [5]. This condition increases bone fragility leading to an increased risk of developing fractures, decreased quality of life, and increased disability and mortality rates and a socioeconomic burden to public health systems [6]. Advancing age, gender (female), race (Asian), early menopause, low body weight, low body mass index (BMI), alcohol and tobacco use, family history of osteoporotic fractures, low intake of calcium or vitamin D, and inadequate sun exposure are the risk factors associated with osteoporosis [7]. 

Inflammatory changes caused by chronic gastritis/ulcers may include disturbed calcium absorption. Low calcium intake is a major risk factor of osteoporosis [8,9,10]. In this regard, PUD may increase the risk of developing osteoporosis, and this condition is commonly associated with gastrointestinal diseases such as celiac disease, chronic liver disease, and inflammatory bowel disease; status after post-gastrectomy, gastric bypass, and liver transplantation; pancreatic insufficiency; and total parenteral nutrition [11]. However, the association between PUD and osteoporosis is not well known. Recently, two studies have explored the relationship between PUD and osteoporosis. Results have shown that PUD increased the risk of developing osteoporosis [12,13]. However, Sawicki et al. had small populations and Wu et al. did not include important confounding factors such as smoking and alcohol, and menopause. This result was not enough to support the association between PUD and osteoporosis. In this regard, we analyzed the relationship between PUD and osteoporosis according to gender for a more accurate analysis. Therefore, we investigated the association between PUD and osteoporosis by classifying individuals based on gender in a prospective study on 10,038 adults in Korea at 12-year follow-up examination.

## 3. Methods

### 3.1. Study Participants

The study was based on data obtained from the Ansung–Ansan cohort studies, which are ongoing prospective studies that are part of the Korean Genome and Epidemiology Study (KoGES). The KoGES started in 2001 and investigated chronic diseases (diabetes, hypertension, osteoporosis, obesity, and metabolic syndrome). The baseline survey of the Ansung–Ansan cohort studies was conducted from May 2001 to February 2003, and 10,030 participants (5018 from the Ansung study and 5012 from the Ansan study) completed the examination. The Ansung and Ansan cohort studies, which consist of Korean men and women aged 40–69 years with the same ethnic background, represent both rural and urban communities, respectively. The details of these cohort studies have been previously described [14]. The participants in the Ansung–Ansan cohort studies measured the speed of sound (SOS, m/s) using quantitative ultrasound (QUS) at their mid-distal radius and tibia of non-dominant arm and leg at baseline, respectively (Omnisense 7000s, Sunlight Medical Ltd., Eilat, Israel). This was measured three times and their average were described as the final value. The SOS was biennially measured until 2008 in the Ansan area and until 2014 in the Ansung area.

The participants who did not answer the questionnaire of experience of diagnosis or treatment of peptic ulcer or currently receiving treatment of peptic ulcer (*n* = 5473), who had not completed the food frequency questionnaire (*n* = 105), who had no data on the T-score of their radius or tibia and body weight (*n* = 368), who were taking drugs for osteoporosis (*n* = 121), who did not answer the questionnaires about drinking, smoking, and menopause (women) (*n* = 53), and who were diagnosed with osteoporosis based on the T-score of their radius and tibia at baseline (*n* = 431), were excluded from the study. Finally, 3479 participants (1653 men and 1826 women) were included (Figure 1). The study protocol was approved by the institutional review board of the Korea Centers for Disease Control and Prevention.

### 3.2. Questionnaires

Data on the participants’ general characteristics, including age, sex, drinking and smoking status, physical activity (PA), hormone replacement therapy, and menopause, were obtained via interview using a questionnaire. Age was stratified as under 50 and over 50 years. The intensity of physical activity was categorized according to the tertile of metabolic equivalent of task values. Data on the amount of calcium intake were obtained using the food frequency questionnaire. Calcium intake levels were divided into quartiles for analysis. Comorbidities, such as hypertension, diabetes mellitus, chronic kidney disease, chronic pulmonary disease, cerebrovascular disease, coronary artery disease, and hepatitis, were included in the analysis.

Data on peptic ulcer and gastritis were obtained using questionnaires. The participants who were diagnosed with peptic ulcer and gastritis and who were currently under treatment or taking medications were considered to have PUD. Others were classified as healthy. Previous studies, including epidemiologic and clinical studies, have shown a strong association between gastritis and peptic ulcers [15,16,17].

### 3.3. Anthropometric Parameters and Bone Mass Density

BMI was calculated by dividing weight in kg by height in m^2^. Adults in Asia were classified as normal and obese based on the cut-off value for BMI as defined by the World Health Organization (WHO): normal weight, 18.5–24.9 kg/m^2^; underweight, <18.5 kg/m^2^; and obese, ≥25.0 kg/m^2^. Obesity was categorized as moderate obesity (BMI: 25.0–29.9 kg/m^2^), severe obesity (BMI: 30–34.9 kg/m^2^), and extremely severe obesity (BMI: ≥35 kg/m^2^). The participants were classified as normal, osteopenic, and osteoporotic based on the T-score criteria by the WHO (normal: T-score ≥ −1.0 in both sites; osteopenia: T-score between −1.0 and −2.5 in at least one site; osteoporosis: T-score ≤ −2.5 in at least one site).

### 3.4. Statistical Analysis

Data were expressed as the mean and standard error (continuous variables) or as frequencies and percentages (categorical variables). The groups were compared using t-test (continuous variables) and chi-square test (categorical variables). Cox proportional hazard models were used to calculate the hazard ratio (HR) and 95% confidence interval (CI). Incidence time was calculated by subtracting the date of the first participation from the date of the last participation. Multivariate analysis was adjusted for calcium intake, age, BMI, physical activity, and comorbidities. Analyses were conducted according to gender because hormone therapy and menopause were only applicable in women. Data analysis was performed using STATA version 15.0 (StataCorp LP, College Station, TX, USA). *p*-values < 0.05 were considered statistically significant.

### 3.5. Patient and Public Involvement

This cohort is managed by the Korea Center for Disease Control and Prevention (KCDC). This cohort included only those who gave written informed consent before participating in the cohort. This cohort data is open source. If a researcher submits a research plan to the KCDC and obtains permission, he or she will receive the data. Data is provided in an encrypted, unidentifiable form.

## 4. Results

### 4.1. Baseline Characteristics of the Participants

The baselines characteristics of the male and female participants according to the presence or absence of peptic ulcers are shown in Table 1 and Table A1. The mean ± standard deviation of age were 52.5 ± 0.6 and 51.7 ± 0.6 years in men and women in the PUD group, respectively. The mean calcium intakes of the PUD group were categorized according to quartile: 213.9 ± 8.1, 373.2 ± 5.5, 508.9 ± 6.8, and 866.1 ± 40.6 in men and 205.6 ± 6.9, 376.1 ± 5.4, 517.4 ± 7.9, and 827.1 ± 41.5 in women. In the PUD group, 63.5% (120/189) of men and 23% (43/187) of women were current drinkers. The participants in the PUD group were more physically active than those in the control group (men: 25.7 ± 1.2 vs. 20.4 ± 0.4, *p* < 0.001; women: 24.8 ± 1.2 vs. 19.9 ± 0.3, *p* < 0.001). The prevalence of diabetes mellitus was significantly higher in the non-PUD group. 

### 4.2. Incidence and HRs of Osteoporosis

Around 11.1% (21/189) of men and 29.9% (56/187) of women in the PUD group developed osteoporosis. In the control group, 4.8% (70/1464) of men and 16.5% (271/1639) of women were presented with osteoporosis. The incidence rates per 1000 person–years were 20.5% and 68.5% in men and women in the PUD group and 11.2% and 42.3% in men and women in the control group, respectively, as presented in Table 2. The risk of developing osteoporosis was higher in both men (HR = 1.72, 95% CI = 1.02–2.92) and women (HR = 1.62, 95% CI = 1.20–2.18) in the PUD group than in the control group. Figure 2 shows the cumulative incidence of osteoporosis in both men and women. The incidence rate and HRs of osteoporosis were adjusted for calcium intake, age, BMI, drinking, smoking, physical activity, and comorbidities (hypertension, diabetes mellitus, chronic kidney disease, chronic pulmonary disease, cerebrovascular disease, coronary artery disease, and hepatitis) in both men and women and for hormone replacement therapy and menopause in women only.

### 4.3. Risk Factors of Osteoporosis

In Table 3, the risk factors of osteoporosis in men were age (over 50 years; HR = 1.82, 95% CI = 1.07–3.09), BMI (30–34.9 kg/m^2^; HR = 3.21, 95% CI = 1.13–9.12), and hepatitis (HR = 2.06, 95% CI = 1.03–4.13). In women, the risk factors of osteoporosis were age (over 50 years; HR = 4.43, 95% CI = 1.85–10.60), BMI (25.0–29.9 kg/m^2^; HR = 1.37, 95% CI = 1.08–1.74), BMI (30–34.9 kg/m^2^; HR = 1.47, 95% CI = 1.00–2.17), high level of physical activity (HR = 1.31, 95% CI = 1.00–1.71), chronic pulmonary disease (HR = 3.10, 95% CI = 1.41–6.83), coronary artery disease (HR = 2.83, 95% CI = 1.36–5.90), and menopause (HR = 2.14, 95% CI = 1.40–3.29). Calcium intake did not increase the risk of developing osteoporosis in all the participants. In the PUD group, the Cox regression analysis was adjusted for calcium intake, age, BMI, drinking, smoking, physical activity, and comorbidities (hypertension, diabetes mellitus, chronic kidney disease, chronic pulmonary disease, cerebrovascular disease, coronary artery disease, and hepatitis) in both men and women, and for hormone replacement therapy and menopause in women only. 

## 5. Discussion

The present study showed that PUD increased the risk of developing osteoporosis in both men and women. Around 11.1% and 4.8% of men in the PUD and control groups developed osteoporosis. Meanwhile, 29.9% and 16.5% of women in the PUD and control groups presented with osteoporosis. In men, approximately 1.72 of the participants were at risk for osteoporosis (95% CI = 1.02–2.92). The risk factors of osteoporosis in men were age (over 50 years, BMI (30–34.9 kg/m^2^) and hepatitis. Approximately 1.62 of the women participants in the PUD group was at risk of developing osteoporosis (95% CI = 1.20–2.18). The risk factors of osteoporosis in women were age (over 50 years), BMI (25.0–34.9 kg/m^2^), high level of physical activity, chronic pulmonary disease, coronary artery disease, and menopause. 

In the present study, the risk of developing osteoporosis differed according to sex. This association may be partly attributed to sex hormones. Men with osteoporosis have a higher prevalence of *H. pylori* infection, increased bone turnover, and reduced estrogen level [18]. Testosterone delays the healing process of ulcers, thus decreasing blood flow at the margin of the ulcers and increasing the proinflammatory cytokines, such as interleukin-1β and tumor necrosis factor-α [19]. The proinflammatory cytokines might have regulated osteoblasts and osteoclasts, and as a result, these cytokines may influence the development of osteoporosis [20]. In this regard, we analyzed the relationship between PUD and osteoporosis according to gender for a more accurate analysis. The association between PUD and osteoporosis remains unclear. Therefore, further research must be conducted to validate these results. PUD may impair the absorption of calcium. Christiansen et al. have shown the interaction of calcium and gastrin in patients with duodenal ulcers [21]. Moreover, calcium influx across or into the cell membrane is a primary factor associated with the stimulation of gastrin, and patients with duodenal ulcers are more sensitive than those without ulcers.

A recent study has reported that calcium intake in women with PUD is similar to that of healthy participants [13]. In this regard, the mean calcium intake of the participants was investigated and categorized into quartile. Contrary to the previous study [13], our study showed that calcium intake might not increase the risk of developing osteoporosis in both men and women (HR: 0.74 [0.42–1.29], 0.55 [0.29–1.03], and 0.78 [0.44–1.38] for calcium intake categorized into quartiles 2, 3, and 4 in men, respectively; HR: 1.02 [0.75–1.38], 1.06 [0.79–1.43], 0.87 [0.62–1.20] for calcium intake categorized into quartiles 2, 3, and 4 in women, respectively). However, the exact association between PUD and calcium intake remains unknown. Therefore, future studies must be conducted to validate these results.

The following are the key strengths of the present study. This is a prospective study with a large cohort of participants, where a 12-year follow-up examination was conducted. In addition, we assessed the association between PUD and osteoporosis according to the sex of the participant. Calcium intake was evaluated by qualified interviewers using questionnaires, and the results were calculated by nutritionists [22]. Nevertheless, this study has some limitations. First, our study lacked information on the major factors of PUD such as *H. pylori* infection or the use of NSAIDs. However, *H. pylori* infection and the long-term use of NSAIDs are not always associated with PUD. Chow et al. have shown that ulcers that is not associated with the use of NSAIDs and *H. pylori* infection accounted for approximately 19% of PUD [23]. In a recent systematic review, proton pump inhibitor (PPI) use might increase fracture risk. However, there was no effect of PPI use on BMD [24]. Second, recall bias may have occurred because the KoGES data were based on questionnaires. To minimize this limitation, we considered PUD as the participants who were diagnosed with peptic ulcer and gastritis and who were currently under treatment or taking medications. Third, calcium intake was considered only at baseline, and not during the follow-up. Fourth, we had no data on bone mineral density (BMD) as measured by dual energy X-ray absorptiometry (DEXA) which is the golden standard for the diagnosis of osteoporosis. We defined the osteoporosis using SOS. However, the SOS highly correlates with DEXA [25].

## 6. Conclusions

In conclusion, the risk of developing osteoporosis was significantly higher in both men and women in the PUD group than in the control group (men: HR = 1.72, 95% CI = 1.02–2.92; women: HR = 1.62, 95% CI = 1.20–2.18). Around 11.1% and 4.8% of men in the PUD and control groups developed osteoporosis. Meanwhile, 29.9% and 16.5% of women in the PUD and control groups were presented with osteoporosis. Future prospective studies must be conducted to elucidate the associations between PUD and osteoporosis.

## Figures and Tables

**Figure 1 ijerph-16-02777-f001:**
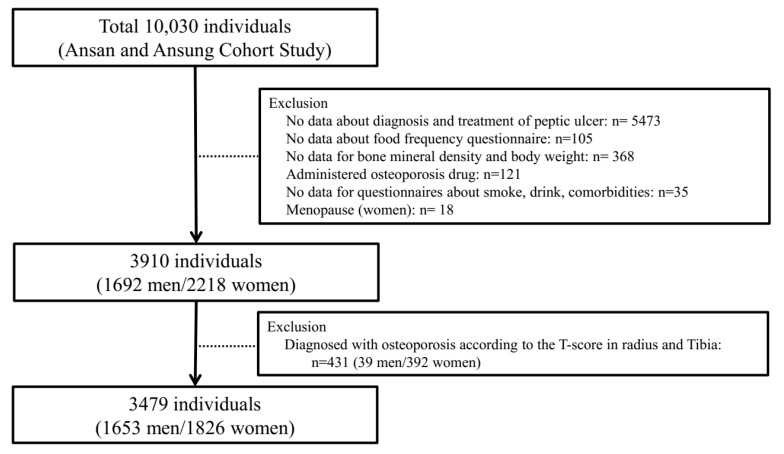
Study population in the present study obtained from the Ansung–Ansan cohort study.

**Figure 2 ijerph-16-02777-f002:**
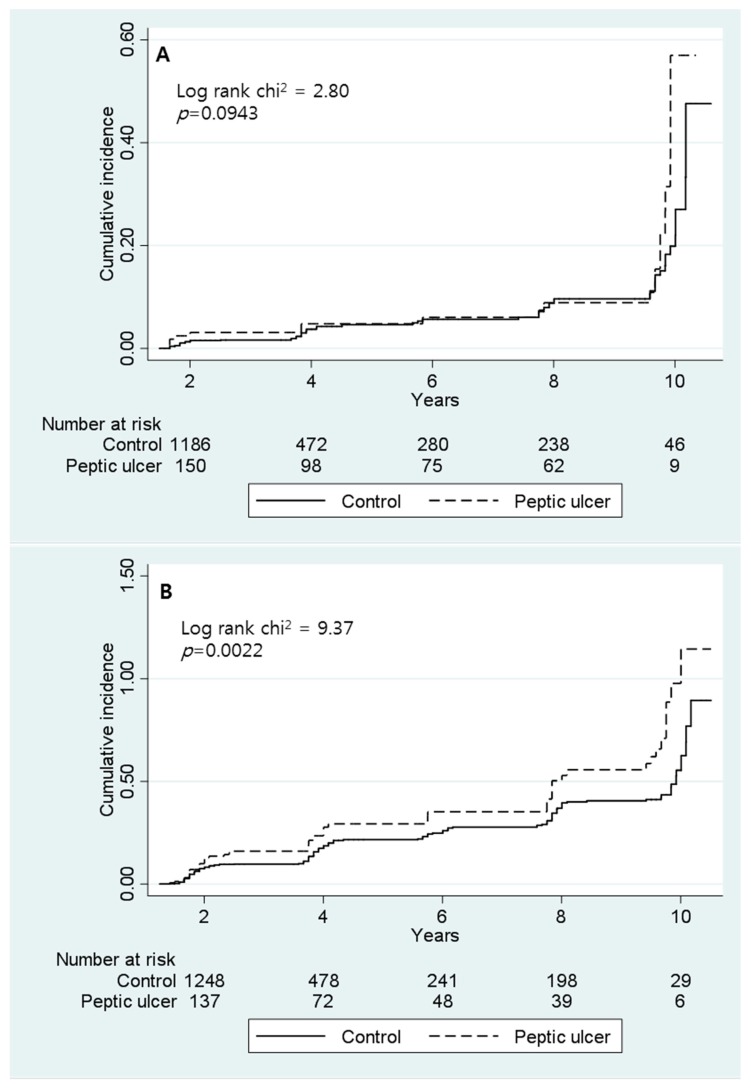
Cumulative incidence of osteoporosis in men (**A**) and women (**B**) with or without peptic ulcer disease.

**Table 1 ijerph-16-02777-t001:** Baseline characteristics in men and women with or without peptic ulcer disease.

Variables	Men		*p*-Value	Women		*p*-Value
	Control	Peptic Ulcer		Control	Peptic Ulcer	
Participants, *n*	1464	189		1639	187	
Calcium intake, mg/d						
Quartile 1	236.1 ± 2.9	213.9 ± 8.1	0.007	213.4 ± 2.5	205.6 ± 6.9	0.269
Quartile 2	375.1 ± 2.0	373.2 ± 5.5	0.742	357.6 ± 2.0	376.1 ± 5.4	0.002
Quartile 3	519.6 ± 2.4	508.9 ± 6.8	0.16	511.2 ± 2.3	517.4 ± 7.9	0.428
Quartile 4	825.1 ± 12.6	866.1 ± 40.6	0.289	809.6 ± 11.9	827.1 ± 41.5	0.658
Age, y	52.1 ± 0.2	52.5 ± 0.6	0.556	51.2 ± 0.2	51.7 ± 0.6	0.475
BMI, kg/m^2^	24.6 ± 0.1	23.4 ± 0.2	<0.001	25.1 ± 0.1	24.1 ± 0.2	<0.001
Drinking Status, *n* (%)			0.007			0.006
Never	295 (20.2)	30 (15.9)		1176 (71.8)	130 (69.5)	
Former	184 (12.6)	39 (20.6)		49 (3.0)	14 (7.5)	
Current	985 (67.3)	120 (63.5)		414 (25.3)	43 (23.0)	
Smoking status, *n* (%)			0.029			0.148
Never	300 (20.5)	25 (13.2)		1570 (95.8)	174 (93.1)	
Former	525 (35.9)	66 (34.9)		14 (0.9)	3 (1.6)	
Current	639 (43.7)	98 (51.9)		55 (3.4)	10 (5.4)	
Physical activity, MET-h/week	20.4 ± 0.4	25.7 ± 1.2	<0.001	19.9 ± 0.3	24.8 ± 1.2	<0.001
Comorbidity, *n* (%)						
Hypertension	370 (25.3)	36 (19.1)	0.061	392 (23.9)	26 (13.9)	0.002
Diabetes mellitus	221 (15.1)	12 (6.4)	0.001	170 (10.4)	6 (3.2)	0.002
Chronic kidney disease	41 (2.8)	3 (1.6)	0.471	62 (3.8)	10 (5.4)	0.298
Chronic pulmonary disease	15 (1.0)	3 (1.6)	0.451	12 (0.7)	-	0.625
Cerebrovascular disease	28 (1.9)	5 (2.7)	0.498	20 (1.2)	1 (0.5)	0.716
Coronary artery disease	27 (1.8)	2 (1.1)	0.766	16 (1.0)	3 (1.6)	0.434
Hepatitis	85 (5.8)	15 (7.9)	0.248	54 (3.3)	10 (5.4)	0.148
Hormone replacement therapy	-	-	-	103 (6.3)	7 (3.7)	0.166
Menopause	-	-	-	941 (57.4)	114 (61.0)	0.352

**Table 2 ijerph-16-02777-t002:** Incidence and hazard ratios of osteoporosis in men and women with or without peptic ulcer disease.

Gender	Group	Osteoporosis	PY ^1^	Rate ^2^	IRR ^3^ (95% CI ^4^)	Adjusted HR ^5,^* (95% CI)
Men	Control (*n* = 1464)	70	6224.0	11.2	1.82 (1.06–3.00)	1.72 (1.02–2.92)
	Peptic ulcer (*n* = 189)	21	1024.6	20.5		
Women	Control (*n* = 1639)	271	6402.7	42.3	1.52 (1.19–2.16)	1.62 (1.20–2.18)
	Peptic ulcer (*n* = 187)	56	817.4	68.5		

^1^ PY: person–years, ^2^ Rate: incidence rate in per 1000 person–years ^3^ IRR: incidence rate ratio in per 1000 person–years, ^4^ 95% CI: 95% confidence interval, ^5^ HR: Hazard ratio. * Adjusted for calcium intake age, BMI, drinking, smoking, physical activity, and comorbidities (hypertension, diabetes mellitus, chronic kidney disease, chronic pulmonary disease, cerebrovascular disease, coronary artery disease, hepatitis) in men. In women, hormone replacement therapy and menopause were also adjusted for, along with these variables. HR and 95% CI were calculated using Cox proportional hazard models.

**Table 3 ijerph-16-02777-t003:** Risk factors of osteoporosis in men and women.

Variables	Men		Women	
	HR *	95% CI	HR ^1,^*	95% CI ^2^
Calcium intake, mg/d				
Quartile 1			1.00	
Quartile 2	0.74	0.42–1.29	1.02	0.75–1.38
Quartile 3	0.55	0.29–1.03	1.06	0.79–1.43
Quartile 4	0.78	0.44–1.38	0.87	0.62–1.20
Age, year				
<50			1.00	
≥50	1.82	1.07–3.09	4.43	1.85–10.60
BMI, kg/m^2^				
18.5–24.9			1.00	
25.0–29.9	1.48	0.95–2.31	1.37	1.08–1.74
30–34.9	3.21	1.13–9.12	1.47	1.00–2.17
>35	-		2.70	0.99–7.41
<18.5	0.94	0.22–3.95	0.97	0.35–2.67
Drinking Status				
Never			1.00	
Former	0.75	0.39–1.44	0.88	0.48–1.62
Current	0.69	0.41–1.17	0.90	0.68–1.19
Smoking status				
Never			1.000	
Former	0.95	0.52–1.72	2.32	0.93–5.78
Current	1.09	0.61–1.95	0.90	0.49–1.67
Physical activity, MET-h/week				
Low			1.00	
Mid	0.77	0.39–1.49	1.02	0.75–1.38
High	1.16	0.68–1.95	1.31	1.00–1.71
Comorbidity				
Hypertension	1.10	0.69–1.75	1.01	0.79–1.29
Diabetes mellitus	1.15	0.67–1.98	1.06	0.77–1.45
Chronic kidney disease	1.58	0.62–4.00	0.67	0.36–1.26
Chronic pulmonary disease	0.52	0.07–3.85	3.10	1.41–6.83
Cerebrovascular disease	-		1.37	0.62–2.99
Coronary artery disease	1.66	0.58–4.77	2.83	1.36–5.90
Hepatitis	2.06	1.03–4.13	0.57	0.27–1.16
Menopause			2.14	1.40–3.29
Hormone replacement therapy			0.72	0.43–1.19

^1^ HR: Hazard ratio, ^2^ 95% CI: 95% confidence interval. * Adjusted for calcium intake age, BMI, drinking, smoking, physical activity, and comorbidities (hypertension, diabetes mellitus, chronic kidney disease, chronic pulmonary disease, cerebrovascular disease, coronary artery disease, hepatitis) in men. In women, hormone replacement therapy and menopause were also adjusted for, along with these variables. HR and 95% CI were calculated using Cox proportional hazard models.

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
