# Peer review of "Association between Peptic Ulcer Disease and Osteoporosis: The Population-Based Longitudinal Cohort Study in Korea"

_ijerph, 2019, doi:10.3390/ijerph16152777_

Round 1

Reviewer 1 Report

The work by Yoon and co-authors concerns the assessment of the relationship between occurrence of  peptic ulcer disease and osteoporosis. The authors in a prospective study of took into account an adequately large group of patients evaluated in a 12-year follow-up. The presented work, despite the constraints indicated at the beginning, is a valuable study in this aspect which is still insufficiently understood and described, but manuscript requires correction of some aspects. Below I present minor comments to the evaluated manuscript.

In my opinion, the introduction section is quite general, weakly introducing the discussed issue and should be enriched.

The aim of this work should be strongly and clearly marked on the and of introduction section.

The discussion contains a significant part of the repetition of the results obtained in this research, and really slight part of real discussion. Similarly, the conclusions, which should be a clear summary of the work, also include repeating the results. There is no discussion of the issue in a wider spectrum and based on the work of other authors, only a few works have been indicated. The presence of only a few studies on the same topic is not the basis for a quick discussion, because many aspects can be its basis.

Author Response

Dear Editor in Chief

We would like to thank you and the reviewers of IJERPH for taking the time to review our manuscript. We have made some corrections and clarifications in the manuscript after going over the reviewers’ comments.

We thank you and the reviewers for your valuable comments and helpful suggestions, which have contributed significantly to the improvement of our manuscript.

We have revised and resubmitted the manuscript based on the review and added the requested data and discussion. These are indicated by the yellow-highlighted text in the revised manuscript.

Reviewer 1>

The work by Yoon and co-authors concerns the assessment of the relationship between occurrence of peptic ulcer disease and osteoporosis. The authors in a prospective study of took into account an adequately large group of patients evaluated in a 12-year follow-up. The presented work, despite the constraints indicated at the beginning, is a valuable study in this aspect which is still insufficiently understood and described, but manuscript requires correction of some aspects. Below I present minor comments to the evaluated manuscript.

1) In my opinion, the introduction section is quite general, weakly introducing the discussed issue and should be enriched. The aim of this work should be strongly and clearly marked on the and of introduction section.

Answer: We appreciate your valuable comment. According to the reviewer’s comments, we have revised the manuscript in this regard (in Introduction section) as below.

“However, Sawicki et al. had small populations and Wu et al. did not include important confounding factors such as smoking and alcohol, and menopause. This result is not enough to support the association between PUD and osteoporosis. In this regard, we analyzed the relationship between PUD and osteoporosis according to gender for a more accurate analysis.” (Introduction section)

2) The discussion contains a significant part of the repetition of the results obtained in this research, and really slight part of real discussion. Similarly, the conclusions, which should be a clear summary of the work, also include repeating the results. There is no discussion of the issue in a wider spectrum and based on the work of other authors, only a few works have been indicated. The presence of only a few studies on the same topic is not the basis for a quick discussion, because many aspects can be its basis

Answer: We appreciate your valuable comment. According to the reviewer’s comments, we have revised the manuscript in this regard (in Result and Discussion section) as below.

“The baselines characteristics of the male and female participants according to the presence or absence of peptic ulcers are shown in Table 1 and supplementary Table 1. The mean±standard deviation of age were 52.5±0.6 and 51.7±0.6 years in men and women in the PUD group, respectively. The mean calcium intakes of the PUD group were categorized according to quartile: 213.9±8.1, 373.2±5.5, 508.9±6.8, and 866.1±40.6 in men and 205.6±6.9, 376.1±5.4, 517.4±7.9, and 827.1±41.5 in women. In the PUD group, 63.5% (120/189) of men and 23% (43/187) of women were current drinkers. The participants in the PUD group were more physically active than those in the control group (men: 25.7±1.2 vs 20.4±0.4 [p <0.001]; women: 24.8±1.2 vs 19.9±0.3 [p <0.001]). The prevalence of diabetes mellitus was significantly higher in the non-PUD group.” (Baseline characteristics of the participants in Introduction section)

“Around 11.1% and 4.8% of men in the PUD and control groups developed osteoporosis. Meanwhile, 29.9% and 16.5% of women in the PUD and control groups presented with osteoporosis. In men, approximately 1.72 of the participants were at risk for osteoporosis (95% CI=1.02–2.92). The risk factors of osteoporosis in men were age (over 50 years, BMI (30–34.9 kg/m2), and hepatitis. Approximately 1.62 of the participants in the PUD group was at risk of developing osteoporosis (95% CI=1.20–2.18) in women.” (Discussion section)

“Contrary to previous study[13], our study showed that calcium intake might not increase the risk of developing osteoporosis in both men and women (HR: 0.74 [0.42–1.29], 0.55 [0.29–1.03], and 0.78 [0.44–1.38] for calcium intake categorized into quartiles 2, 3, and 4 in men, respectively.” (Discussion section)

“In addition, we assessed the association between PUD and osteoporosis according to the sex.” (Discussion section)

We would like to confirm again that there is nothing to be declared and all authors have approved the revised manuscript. We hope that our revised manuscript will better meet the requirement of IJERPH for publication. And we thank you for valuable comments by reviewers.

Very sincerely yours,

Seok-Hoo Jeong and Yun-Jung Yang

Reviewer 2 Report

Yoon et al investigated association between peptic ulcer disease (PUD) and osteoporosis in a prospective study based on the Ansung-Ansan cohort studies in Korea. The risk of osteoporosis was found to be higher in the PUD group compared with the control group. The study is straightforward, and the manuscript is well written. However, there are rooms for improvement in this manuscript.

1. Line 85. Speed of sound (SOS) is poorly explained in the text. The authors should explicitly introduce SOS as ultrasonic measurement. Which part of bone was used for the measurement?

2. Exclusion criteria sound ambiguous. “who did not complete the questionnaire about the diagnosis OR treatment of peptic ulcer (Line 87)” is different from “No data about diagnosis AND treatment of peptic ulcer”.  

3. It is unclear why the authors excluded participants “who were diagnosed with osteoporosis based on the T-score of their radius and tibia at baseline (n=431) were excluded from the study (Lines 91-92)”. Please explain.

4. Figure 2. The top panel B should be labeled panel A, and the bottom panel A should be labeled panel B. “Number at risk” should be explained in detail in the legend.

5. Lines 139-142. The text just describes the values shown in Table 1. The authors should improve the description not to be repetitive.

6. “Calcium intake did not increase the risk of developing osteoporosis (Line 35, 180)” sounds odd considering the fact that “Low calcium intake is a major risk factor of osteoporosis (Line 64)”. The same is true for “Our study showed that calcium intake might not increase the risk of developing osteoporosis (Line 216)”.

Author Response

July 31, 2019

Dear Editor in Chief

We would like to thank you and the reviewers of IJERPH for taking the time to review our manuscript. We have made some corrections and clarifications in the manuscript after going over the reviewers’ comments.

We thank you and the reviewers for your valuable comments and helpful suggestions, which have contributed significantly to the improvement of our manuscript.

We have revised and resubmitted the manuscript based on the review and added the requested data and discussion. These are indicated by the yellow-highlighted text in the revised manuscript.

Reviewer 2>

Yoon et al investigated association between peptic ulcer disease (PUD) and osteoporosis in a prospective study based on the Ansung-Ansan cohort studies in Korea. The risk of osteoporosis was found to be higher in the PUD group compared with the control group. The study is straightforward, and the manuscript is well written. However, there are rooms for improvement in this manuscript.

Line 85. Speed of sound (SOS) is poorly explained in the text. The authors should explicitly introduce SOS as ultrasonic measurement. Which part of bone was used for the measurement?

Answer: We appreciate your valuable comment. According to the reviewer’s comments, we have revised the manuscript in this regard (in Introduction section) as below.

“The participants in the Ansung–Ansan cohort studies measured the speed of sound (SOS, m/s) using quantitative ultrasound (QUS) at their mid-distal radius and tibia of non-dominant arm and leg at baseline, respectively (Omnisense 7000s, Sunlight Medical Ltd. Israel). This was measured three times and their average were described as the final value.” (Study participants in Method section)

Exclusion criteria sound ambiguous. “who did not complete the questionnaire about the diagnosis OR treatment of peptic ulcer (Line 87)” is different from “No data about diagnosis AND treatment of peptic ulcer”.  

Answer: We appreciate your valuable comment. According to the reviewer’s comments, we have revised the manuscript in this regard (in Introduction section) as below.

“The participants who did not answer the questionnaire of experience of diagnosis or treatment of peptic ulcer or currently receiving treatment of peptic ulcer (n=5,473),” (Study participants in Method section)

It is unclear why the authors excluded participants “who were diagnosed with osteoporosis based on the T-score of their radius and tibia at baseline (n=431) were excluded from the study (Lines 91-92)”. Please explain.

Answer: We appreciate your valuable comment. This study was performed to analysis the association between peptic ulcer and the development of osteoporosis. If people who are classified as having osteoporosis in the baseline study are included, it is difficult to determine the causal association between PUD and osteoporosis. We classified osteoporosis when the T score of radius or tibia was –2.5 or less. The score was determined according to the WHO guideline (line 117-120). So, those who met this range (n=431) were excluded from the analysis.”

Figure 2. The top panel B should be labeled panel A, and the bottom panel A should be labeled panel B. “Number at risk” should be explained in detail in the legend.

Answer: We appreciate your valuable comment. According to the reviewer’s comments, we have revised the order as below. The ‘number at risk’ represents the number of subjects at the time of observation.”

Lines 139-142. The text just describes the values shown in Table 1. The authors should improve the description not to be repetitive.

Answer: We appreciate your valuable comment. According to the reviewer’s comments, we have revised the manuscript in this regard (in Result and Discussion section) as below.

“The baselines characteristics of the male and female participants according to the presence or absence of peptic ulcers are shown in Table 1 and supplementary Table 1. The mean±standard deviation of age were 52.5±0.6 and 51.7±0.6 years in men and women in the PUD group, respectively. The mean calcium intakes of the PUD group were categorized according to quartile: 213.9±8.1, 373.2±5.5, 508.9±6.8, and 866.1±40.6 in men and 205.6±6.9, 376.1±5.4, 517.4±7.9, and 827.1±41.5 in women. In the PUD group, 63.5% (120/189) of men and 23% (43/187) of women were current drinkers. The participants in the PUD group were more physically active than those in the control group (men: 25.7±1.2 vs 20.4±0.4 [p <0.001]; women: 24.8±1.2 vs 19.9±0.3 [p <0.001]). The prevalence of diabetes mellitus was significantly higher in the non-PUD group.” (Baseline characteristics of the participants in Introduction section)

“Calcium intake did not increase the risk of developing osteoporosis (Line 35, 180)” sounds odd considering the fact that “Low calcium intake is a major risk factor of osteoporosis (Line 64)”. The same is true for “Our study showed that calcium intake might not increase the risk of developing osteoporosis (Line 216)”.

Answer: We appreciate your valuable comment. According to the reviewer’s comments, we have revised the manuscript in this regard (Discussion section) as below. In addition, we removed line 35 in Abstract section to focus our major results.

“Contrary to previous study[13], our study showed that calcium intake might not increase the risk of developing osteoporosis in both men and women (HR: 0.74 [0.42–1.29], 0.55 [0.29–1.03], and 0.78 [0.44–1.38] for calcium intake categorized into quartiles 2, 3, and 4 in men, respectively;” (Discussion section)

We would like to confirm again that there is nothing to be declared and all authors have approved the revised manuscript. We hope that our revised manuscript will better meet the requirement of IJERPH for publication. And we thank you for valuable comments by reviewers.

Very sincerely yours,

Seok-Hoo Jeong and Yun-Jung Yang